# The Challenges and Opportunities of the Implementation of Comprehensive Genomic Profiling in Everyday Clinical Practice with Non-Small Cell Lung Cancer: National Results from Croatia

**DOI:** 10.3390/cancers15133395

**Published:** 2023-06-28

**Authors:** Dora Čerina, Kristina Krpina, Marko Jakopović, Natalija Dedić Plavetić, Fran Seiwerth, Snježana Tomić, Jasna Radić, Ingrid Belac Lovasić, Ivana Canjko, Marijo Boban, Miroslav Samaržija, Eduard Vrdoljak

**Affiliations:** 1Department of Oncology, University Hospital Center Split, 21000 Split, Croatia; dora09cerina@gmail.com (D.Č.); marijo.boban@gmx.com (M.B.); 2School of Medicine, University of Split, 21000 Split, Croatia; st@mefst.hr; 3Department for Respiratory Diseases Jordanovac, University Hospital Center Zagreb, 10000 Zagreb, Croatia; krpinakristina@gmail.com (K.K.); marko.jakopovic@kbc-zagreb.hr (M.J.); fseiwerth@gmail.com (F.S.); 4School of Medicine, University of Zagreb, 10000 Zagreb, Croatia; ndedic@kbc-zagreb.hr; 5Department of Oncology, University Hospital Center Zagreb, 10000 Zagreb, Croatia; 6Department of Pathology, Citology and Forensic Medicine, University Hospital Center Split, 21000 Split, Croatia; 7Department of Oncology and Nuclear Medicine, Division of Medical Oncology, University Hospital for Tumors, Sestre Milosrdnice University Hospital Center, 10000 Zagreb, Croatia; jasnaradic@yahoo.com; 8School of Medicine, University of Rijeka, 51000 Rijeka, Croatia; iblovasic@gmail.com; 9Department of Radiotherapy Oncology, University Hospital Center Osijek, 31000 Osijek, Croatia; dr.ivanamarkovic@gmail.com

**Keywords:** comprehensive genomic profiling, NSCLC, precision medicine

## Abstract

**Simple Summary:**

Precision medicine has reached its current peak in non-small cell lung cancer (NSCLC), with a constantly growing number of predictive biomarkers and new targeted therapies that when applied, significantly affect and change outcomes. Hence, the matter in question is how we might optimally detect and implement them in the treatment of our patients in everyday clinical practice. The main problem in the diagnostic workup of NSCLC is the rather limited tumor sample used on many occasions in the classical diagnostic approach, which consists of a series of single-biomarker tests. Consequently, the introduction of comprehensive genomic profiling (CGP) in everyday diagnostic and clinical practice is one of the imperatives that could benefit everybody involved. Here, we present national data and our experiences with the application of comprehensive genomic profiling in NSCLC. The results have shown the utility and potential benefit of comprehensive genomic profiling, but also challenges involved in the implementation of precision oncology in clinical practice. So when possible, CGP should be used as an upfront backbone diagnostic and treatment-oriented work-up in patients with NSCLC.

**Abstract:**

Non–small cell lung cancer (NSCLC) has become the best example of precision oncology’s impact on outcomes in everyday clinical practice, significantly changing the expectations of all stakeholders, including medical professionals, society, and most importantly, patients. Consequently, the implementation of the precision oncology concept in medical systems, in order to achieve optimal and proven curative effects in NSCLC, is imperative. In this study, we investigated the development, challenges, and results associated with the implementation of precision oncology in NSCLC on a national level in Croatia. We conducted a multicenter, retrospective, cross-sectional analysis on the total population of Croatian patients with metastatic lung cancer, on whose tumors specimen comprehensive genomic profiling (CGP) testing was performed during 2020 and 2021. A total of 48 patients were included in the study. CGP revealed clinically relevant genomic alterations (CRGA) in 37 patients (79%), with a median of 2 (IQR 1–3) CRGA per patient. From the panel of recommended tests, *KRAS*, *MET*, and *EGFR* were the most common alterations, detected in 16 (34%), 5 (11%), and 3 (6%) patients, respectively. CGP revealed additional targetable mutations in 29 (60%) patients who would not have been tested (and consequently, whose mutations would not have been detected) according to the existing everyday standard of practice in Croatia. The tumor mutational burden was reported as high (≥10 Muts/Mb) in 19 patients (40%). CGP analysis reported some kind of targeted therapy for 34 patients (72%). CGP revealed other potentially targetable mutations, and it also determined TMB to be high in a significant number of patients. In conclusion, when possible, CGP should be used as an upfront backbone diagnostic and treatment-oriented work-up in patients with NSCLC.

## 1. Introduction

Lung cancer is the leading cause of cancer morbidity and mortality, accounting for approximately 18% of all cancer-related deaths [1]. In Croatia, the burden of lung cancer is rather large, with approximately 3000 newly diagnosed patients per year, making it second most common both in males and females, with a mortality-to-incidence ratio of 0.95 for males and 0.85 for females [2]. Non–small cell lung cancer (NSCLC) is the most common histological subtype, accounting for 85% of all diagnosed lung cancers [3]. Recently, we have witnessed an explosion in the number of predictive biomarkers and aligned targeted therapies that alter the course of treatment, and these have changed outcomes more significantly in NSCLC than in the majority of other cancers [4]. Currently, the National Comprehensive Cancer Network (NCCN) guidelines recommend testing NSCLC for several *EGFR* (epidermal growth factor receptor) mutations, the *KRAS* (Kirsten rat sarcoma virus) G12C mutation, rearrangements of the *ALK* (anaplastic lymphoma kinase) gene, *ROS1* (ROS Proto-Oncogene 1) and *RET* (Ret Proto-Oncogene) mutations, the *BRAF* (B-Raf proto-oncogene) V600E and *ERBB2* (erb-b2 receptor tyrosine kinase 2) mutations, *NTRK* (neurotrophic tyrosine receptor kinase) gene fusions, the *MET* (MET Proto-Oncogene) exon 14 skipping mutation, and for determination of PD-L1 (programmed death-ligand 1) status [5]. The European Society for Medical Oncology (ESMO) recommends testing for same alterations as NCCN, but with only EGFR mutations and ALK rearrangements as ESCAT (ESMO Scale of Clinical Actionability for molecular Targets) level I-A [6]. This expansion in biomarkers should be accompanied by appropriate tools to detect them in the fastest, most precise and most cost-effective way. Recently, a major shift has occurred in medicine, particularly in oncology, with the development of targeted therapy and immunotherapy and the tailoring of treatment for every individual patient. One of the contributors to the new era of precision medicine was the evolution of novel technologies such as next-generation sequencing (NGS). There are different scopes of NGS, from covering only areas of interest and small gene panels to covering hundreds of cancer-related genes and enabling comprehensive genomic profiling (CGP) [7]. CGP provides direct insight into tumor specifics, and is becoming widely applicable in everyday clinical practice. Hence, CGP detects all the currently recommended biomarkers for NSCLC, as well as other potentially targetable alterations, and provides valuable information regarding tumor mutational burden and microsatellite status, all of which help to predict a patient’s response to targeted therapies [8,9,10]. Tissue CGP is not always applicable in NSCLC due to usually relatively limited tumor samples. This is also a problem faced in single-biomarker testing, due to the growing number of biomarkers; thus, liquid CGP analysis from circulating free DNA has shown noninferiority to tissue analysis, and is potentially a defined optimal diagnostic tool for certain patients [11,12]. However, negative findings upon liquid testing do not exclude the presence of a viable tumor with potentially targetable mutations [13].

CGP testing is available in Croatia from both tissue and blood, provided by Foundation Medicine Inc., for patients diagnosed with metastatic disease as a part of the national project for development and implementation of precision oncology, signed in 2019 [14].

As previously stated, the determination of biomarkers in NSCLC guides treatment choices with tyrosine kinase inhibitors (TKIs) and immune checkpoint inhibitors (ICIs), surpassing conservative chemotherapy in survival outcomes and tolerability, and becoming the new standard of care, even across treatment lines [15,16,17,18,19,20,21,22,23,24]. Furthermore, anti-HER2 therapy is now recommended in subsequent lines [25,26]. Consequently, NSCLC has become the best example of precision oncology’s impact on outcomes in everyday clinical practice, significantly changing the expectations of all stakeholders, including medical professionals, society, and most importantly, patients. Therefore, the implementation of precision oncology in worldwide medical systems to achieve optimal and proven outcome effects in NSCLC patients is imperative. However, questions remain; should we use limited tumor samples on many occasions for a series of single tests, or should we start with CGP? What are the real-life problems encountered when implementing precision oncology? In this study, we have investigated the development, challenges, and results associated with the implementation of precision oncology in NSCLC on a national level in Croatia, and here, we present our real-world data of CGP analysis in NSCLC.

## 2. Materials and Methods

### 2.1. Project Design

The study was cross-sectional and retrospective in nature; it was conducted in multiple centers and included the total population of Croatian patients who were diagnosed with metastatic NSCLC from 1 January 2020 to 31 December 2021, and whose tumors underwent CGP analysis. The analysis of tumor tissue or blood samples was performed through FoundationOneCDx or FoundationOneLiquidCDx for all patients in a certified Clinical Laboratory Improvement Amendments, College of American Pathologists-accredited laboratory (Foundation Medicine Inc., FMI, Cambridge, MA, USA) [27,28,29,30]. The obtained tumor specimen was sampled from a surgical resection or biopsy of the primary disease or metastases, and for liquid analysis, an anticoagulated peripheral blood sample was obtained.

This real-world analysis was conducted in six Croatian institutions: University Hospital Centre Split, University Hospital Center Zagreb, Department of Oncology and Nuclear Medicine Sestre Milosrdnice, Lung Disease Clinic “Jordanovac”, and University Hospital Centers Rijeka and Osijek. Informed consent was obtained from all patients for the CGP analysis and data collection. The data file was anonymized before the analysis, and the project was performed in accordance with the World Medical Association Declaration of Helsinki of 1975, as revised in 2013 [31].

In accordance with the journal’s guidelines, we have submitted detailed CGP results as a Appendix A. Additionally, we will provide the rest of our data for the reproducibility of this study in other centers if requested.

### 2.2. Comprehensive Genomic Profiling Analysis

In the case of tissue analysis, formalin-fixed, paraffin-embedded tissue was sent as a block, alongside one hematoxylin and eosin-stained slide or ten unstained slides with one hematoxylin and eosin-stained slide. The minimum surface area was 25 mm^2,^ and the minimum tumor content was 20%, while the optimal surface area of the tumor nuclei was 30%, defined as the number of tumor cells divided by the total number of all cells with nuclei. Once the DNA was extracted, 50–1000 ng underwent whole-genome shotgun library construction and hybridization-based capture to detect alterations in 324 genes in total, including 309 exons related to tumors, 1 promoter region, 1 noncoding RNA and certain regions of introns in 34 frequently rearranged tumor genes. Illumina^®^ HiSeq 4000 was used to sequence hybrid capture-selected libraries to high uniform depth. The typical median depth of coverage was >500×, with >99% of exons at coverage >100×. The sequenced regions were analyzed for four different types of alterations: base substitution, deletion or insertion, copy number variation, and gene redistribution, in a group of genes associated with tumor development. The microsatellite status was based on genome-wide analysis of 95 microsatellite loci, while the tumor mutational burden (TMB) was determined by counting all synonymous and nonsynonymous variants present at 5% allele frequency or greater; the total number was presented as mutations per megabase (Muts/Mb) unit [29].

FoundationOne Liquid CDx is NGS-based in vitro diagnostic tool that utilizes circulating cell-free DNA (cfDNA) isolated from plasma derived from anticoagulated whole blood previously collected in a FoundationOne Liquid CDx Blood Sample Collection Kit [30]. The assay analyzes 324 genes in total, similar to tissue CGP, and it detects substitutions, indels, genomic rearrangements, copy number alterations (CNAs; amplifications and losses), and genomic signatures, including the blood tumor mutation burden (bTMB), microsatellite instability (MSI), and tumor fraction(TF) [12]. Novel high-throughput hybridization capture technology enables the baiting of a subset of targeted regions in 75 genes for greater sensitivity through ultra-deep sequencing coverage (referred to as the enhanced sensitivity region). Additionally, the test is used mainly as a companion diagnostic for NSCLC, prostate, breast and ovarian cancer, and negative results do not indicate that the tumor is negative for genomic alterations; thus, this should be confirmed with tissue analysis [30]. However, FoundationOne Liquid CDx has been approved by the FDA for pan-cancer cfDNA-based CGP.

CGP analysis presents the genomic alterations in two groups: clinically relevant (associated with either on-label or off-label targeted therapy, or a potential clinical trial) and genomic alterations of unknown significance.

PD-L1 status was determined by a VENTANA PD-L1 (SP263) qualitative immunohistochemical assay, which used rabbit monoclonal anti-PD-L1 clone SP263 in formalin-fixed, paraffin-embedded (FFPE) NSCLC tissue stained with an OptiView DAB IHC Detection Kit on a VENTANA BenchMark ULTRA instrument.

### 2.3. Participants

We included the entire population of patients diagnosed with metastatic NSCLC who fulfilled the CGP criteria defined by the Croatian Oncology Society, and whose tumor specimens were tested. The criteria for CGP were good general health (ECOG performance status of 0 or 1), the ability to receive standard systemic treatment and potentially CGP-guided treatment, and at least 6 months of life expectancy [14]. The power analysis was not performed before starting the project. Patients were administered the first-line standard of care treatment for metastatic NSCLC, and were potentially administered CGP-guided therapy in accordance with the clinical assessment, multidisciplinary team decision, and availability of the on-label or off-label drugs in Croatia.

### 2.4. Endpoints

The aim of this study was to address all the challenges and opportunities of the implementation of CGP in the everyday management of metastatic NSCLC through the presentation of our results.

The secondary endpoint was to present our clinical experience and results of the administered CGP-guided therapy.

### 2.5. Statistical Analysis

We described the data as percentages, medians and interquartile ranges (IQRs) using StataCorp 2019 software (Stata Statistical Software: Release 16. College Station, TX, USA: StataCorp LLC).

## 3. Results

### 3.1. Demographic Characteristics of Patients

From 1 January 2020 to 31 December 2021, there were 48 patients diagnosed with metastatic NSCLC who presented to multidisciplinary teams and on whose tumors CGP testing was performed. For 45 patients (94%), analysis was carried out using FoundationOneCDx, while for only 3 patients (6%), it was carried out through FoundationOne Liquid CDx. There was an equal distribution between sexes, with 25 (52%) female and 23 (48%) male patients. The median age at the time of metastatic disease diagnosis was 62 years (IQR 51.5–68 years), and 30 (63%) patients were newly diagnosed with metastatic disease. The majority of patients (37; 77%) received some kind of systemic therapy prior to testing, with a median number of two lines of treatment (IQR 1–2), and the most common treatment was a combination of platinum-based chemotherapy with pemetrexed (in 81% of patients). Before the CGP analysis, 24 (50%) and 22 (46%) patients had ECOG performance statuses of 0 and 1, respectively (Table 1).

### 3.2. Results of the CGP Testing

CGP revealed that the vast majority of patients (96%) had at least one genomic alteration. Clinically relevant genomic alterations (CRGA) were reported in 38 patients (79%), with a median of 2 (IQR 1–3) CRGA per patient (Table 2). From the NCCN recommended biomarker panel, *KRAS* was the most common gene, altered in 16 (33%) patients, out of which the G12C mutation was detected in 4 (25%) patients. Furthermore, *MET* and *EGFR* were the next two most common genes with alterations detected in five (10%) and three (6%) patients, respectively. Other alterations detected from the panel were found in *ROS1*, *ERBB2* and *RET* in 2 (4%), 2 (4%) and 1 (2%) patients, respectively. The next most common CRGAs were found in *STK11* (serine/threonine kinase 11), *KEAP* (Kelch-like ECH-associated protein) and *CTNNB1* (Catenin Beta 1) in 15 (31%), 6 (13%) and 5 (10%) patients, respectively (Table 2). Microsatellite status was determined to be stable in 44 patients (92%), and it could not be determined in 4 patients (8%). TMB was reported as high (≥10 Muts/Mb) in 19 patients (40%). PD-L1 status was determined for 30 (63%) patients, out of which clinically positive results were reported for 23 (77%) patients. Low positive status (1–24%) was reported in 13 (43%) patients, moderately positive (25–49%) in 3 (10%) patients, and highly positive (≥50%) in 7 (23%) patients. A combined high TMB and positive PD-L1 status was found in three patients (6%). CGP revealed additional targetable mutations in 29 (60%) patients who would not have been tested (and consequently, whose mutations would not have been detected) according to the existing everyday standard of practice in Croatia. A detailed presentation of the CGP results is shown in the Appendix A.

### 3.3. CGP-Guided Therapy

After CGP analysis, some kind of targeted therapy was reported for 35 patients (73%). Targeted therapy approved for the patients’ tumor type (on-label therapy) was reported in 27 (56%) patients, while targeted therapy approved in other tumor types based on the patients’ genomic alterations (off-label therapy) was reported in 31 (65%) patients. The vast majority of on-label alteration-driven therapies consisted of ICIs and TKIs. Additionally, the most common off-label therapy was ICIs. In addition, other common alteration-driven off-label therapies included PARP inhibitors involved in DNA repair mechanisms, such as olaparib, niraparib, rucaparib, the mTOR inhibitors everolimus and temsirolimus, CDK4/6 inhibitors, and HER2 inhibitors.

CGP-guided targeted therapy was administered to 13 (27%) patients based upon the indication, clinical assessment, multidisciplinary team decision, and reimbursement status of the therapy or availability. From this group of treated patients, seven (54%) patients were administered targeted therapy according to the CGP, which was not reimbursed in Croatia. Considering the number of previous lines of therapy, a clinically significant duration of CGP-guided treatment was observed in six (46%) patients, with three (23%) patients still receiving the treatment (Table 3). The median progression-free survival was 5 months (IQR 2–17).

## 4. Discussion

Precision medicine has reached a peak with regard to NSCLC, with a constantly growing number of predictive biomarkers and new targeted therapies, and consequently, significantly improved outcomes. Today, NSCLC is divided into molecular subgroups, and diagnostic biomarker testing is of the utmost importance for creating an optimal treatment strategy [4]. However, the availability of comprehensive biomarker testing and the consequent reimbursement of matching drugs are not equally distributed, and their uptake in everyday clinical practice is insufficiently utilized, leaving many patients with NSCLC underserved [32]. Furthermore, the insufficiency of tissue availability, the cost of CGP in some health care systems, and the long turnaround time of testing are also challenges that we face in everyday clinical work [33]. With regard to the size of tissue samples, CGP is the test of interest, as it addresses this even beyond all recommendations; it also provides information about available clinical trials, and its utility has already been proven in several trials [34,35,36,37]. Moreover, NGS has a higher sensitivity than classical polymerase chain reaction (PCR) or immunohistochemistry (IHC), which could lead to a lower chance of omissions or false-negative findings [38,39].

Nevertheless, if not at the time of diagnosis, the question of tissue availability is also a challenge at the time of progression, as we need to detect resistance alterations that are also actionable or potentially novel targets. In those cases, as well as in the case of an initial insufficiency of tissue, liquid biopsy and an analysis of the circulating tumor DNA from the blood is proficient in the detection of alterations, and in some instances less expensive, less invasive, and carrying less risk of complications than tissue rebiopsy [32,40]. Particularly important is its clinical validity and noninferiority in detecting potentially new targets or mechanisms of resistance in patients undergoing different specific therapies [41,42].

Proportional to the increasing number of biomarkers is an increase in cost, as well as a longer time for treatment initiation due to the extended time needed for single-biomarker testing. Even with broader testing methods such as CGP, NGS is often associated with a shorter time to the initiation of the treatment, and with cost reductions in comparison to single-gene assays [43,44].

Our results have shown that the vast majority of patients had at least one genomic alteration, that among more than half of our patients, CGP detected additional targetable mutations outside of the reimbursed testing that we offer, and that more than half of patients also opted for targeted therapy, indicating that CGP is a valuable asset that Croatia has at its disposal. Furthermore, CGP revealed additional targetable mutations in 29 (60%) patients who would not have been tested (and consequently, whose mutations would not have been detected) according to the existing everyday standard of practice in Croatia. Additionally, CGP revealed a high TMB in 40% of patients, which could potentially be beneficial for predicting responses to ICIs. Consequently, to be more precise in the implementation of precision medicine, we have to address all potential insufficiencies of the local diagnostic procedures. In terms of determining biomarkers for potential response to ICI, we can see that TMB overlaps with PD-L1 positivity in only three patients and that all patients had a stable microsatellite status, which means that in our population, TMB is potentially a beneficial biomarker. However, when speaking in terms of cost-effectiveness, at this moment, single testing is still potentially less expensive than CGP, simply because we do not have reimbursed drugs for all recommended biomarkers. Currently, in Croatia, we have reimbursement for *EGFR*, *ALK*, *ROS1* and *NTRK* inhibitors, as well as immunotherapy for NSCLC. Notably, we do not have *BRAF*, *KRAS*, *MET* and HER2-directed therapy available. The question that every oncologist and patient asks is ‘what can we do when we have positive test results, and how can we face the potential lack of opportunity for a specific therapy that could significantly help?’ This could be the reason that some oncologists do not test tumors for biomarkers for which they do not have a therapy reimbursed. On the other hand, diagnosing specific biomarkers is the only hope for a longer and better life for patients with metastatic NSCLC. In addition, 50% of treated patients, in accordance with the CGP findings, responded to targeted therapy that was not reimbursed in Croatia, which gives a glimpse of what CGP could actually bring to our patients. Our clear understanding of this issue is that patients have a basic human right to know the biomarker status of their tumor, and consequently, they should be in a position to try and find a way to receive optimal treatment for their disease.

The major limitation of our study is the relatively small number of patients enrolled and tested, most likely because of either tissue unavailability, longer turnaround time for CGP, or unequal penetration of CGP in all institutions, as well as the above-stated question of the drug availability for detected biomarkers that are not reimbursed in Croatia. Long turnaround time is not based on FMI performance, but on additional administration introduced by the Croatian healthcare system. Based on that, the vast majority of patients have been directed to classical and reimbursed biomarker testing and not to CGP. Thus, we are facing the same issues regarding NSCLC management as have been previously emphasized. However, with appropriate guidelines, the creation of standardized operating procedures and a consequently faster turnaround time for CGP, and with already established specialties found for the treatment of tested patients with biomarker end efficacy-driven and proven targeted drugs, we believe that uptake will be significantly higher, and that we will contribute to the positioning and affirmation of precision medicine for every patient with NSCLC. Of course, all precision oncology programs must be closely monitored, and the results must be reported. Finally, we have to learn from every patient tested and treated, according to the CGP findings. This should be the new paradigm of clinical science in precision oncology.

## 5. Conclusions

CGP revealed other potentially targetable mutations, and it also determined TMB to be high in a significant number of patients. The implementation of precision oncology in different health care systems, particularly in transitional countries such as Croatia, is facing many administrative and financial issues. In conclusion, when possible, CGP should be used as a cornerstone of diagnostic and treatment-oriented work-up in patients with NSCLC.

## Figures and Tables

**Table 1 cancers-15-03395-t001:** Demographic characteristics of patients tested with CGP.

	All Patients N (%)	
FoundationOneCDx	45	(94)
FoundationOne Liquid CDx	3	(6)
Sex		
Male	23	(48)
Female	25	(52)
Age at the time of diagnosis, median (IQR)	62	(51.5–68)
Metastatic disease at the initial diagnosis	30	(63)
Disease progression	15	(31)
Stage of the disease not determined	3	(6)
Number of patients receiving previous chemotherapy	37	(77)
Number of previous treatment lines for metastatic disease		
0	5	(10)
1	36	(75)
2	17	(35)
3	7	(15)
Not determined	3	(6)
ECOG performance status before CGP *		
0	24	(50)
1	22	(46)
Not determined	2	(4)

Data are presented as the numbers (percentages) of patients if not stated otherwise. Abbreviations: IQR, interquartile range; CGP, comprehensive genomic profiling. Data were missing for the date of metastatic disease and number of previous treatment lines for metastatic disease in two (4%) patients. * CGP was performed upon progression.

**Table 2 cancers-15-03395-t002:** Comprehensive genomic profiling results.

	All Patients N (%)	
Genomic alterations (N,%)		
Any genomic alteration	46	(96)
Clinically relevant	38	(79)
Not clinically relevant	41	(85)
Number of genomic alterations, median (IQR)		
Clinically relevant	2	(1–3)
Not clinically relevant	3	(2–4)
Clinically relevant genomic alterations (N,%)		
*KRAS*	16	(33)
*STK11*	15	(31)
*KEAP*	6	(13)
*MET*	5	(10)
*CTNNB1*	5	(10)
*EGFR*	3	(6)
*BRAF*	3	(6)
*ROS1, ERBB2, RICTOR, MYC, ATM,*	2	(4)
*CDK4, MDM2, CHEK2, MTAP*		
*RET, BRCA1, PIK3CA, SMARCBI,*	1	(2)
*ERRFI1, MTAT, AKT2, AXL, CBL,*		
*CDKN1A, NF1*		
PD-L1 status †	30	(63)
Negative	7	(23)
Low positive (1–24%)	13	(43)
Moderately positive (25–49%)	3	(10)
Highly positive (≥50%)	7	(23)
Not determined	18	(38)
Microsatellite status		
Stable	44	(92)
High instability	0	(0)
Not determined	4	(8)
Tumor mutational burden (TMB), median (IQR)	8	(4–13)
Tumor mutational burden (TMB)		
Not high (<10 mutations/Mb)	25	(52)
High (≥10 mutations/Mb)	19	(40)
Not determined	4	(8)

Data are presented as the numbers (percentages) of patients if not stated otherwise. Abbreviations: IQR, interquartile range. † The total is <100%, due to a rounding error.

**Table 3 cancers-15-03395-t003:** CGP-guided therapy.

CRGA	Targeted Therapy	Treatment Duration Months (Outcome)
*BRAF*	Trametinib	2 (DEATH)
*ROS1*	Crizotinib	30 (ONGOING)
High TMB	Atezolizumab	4 (DEATH)
*KRAS*	Sotorasib *	16 (PROGRESSION)
High TMB	Atezolizumab	3 (DEATH)
*RET*	Pralsetinib *	27 (ONGOING)
*EGFR*	Erlotinib	29 (ONGOING)
*MET*	Crizotinib	5 (PROGRESSION)
*MET*	Crizotinib	18 (PROGRESSION)
High TMB	Atezolizumab	1 (DEATH)
*ROS1*	Crizotinib	9 (PROGRESSION)
High TMB	Atezolizumab	1 (DEATH)
*EGFR*	Osimertinib	2 (DEATH)

Abbreviations: CRGA, clinically relevant genomic alteration; CGP, comprehensive genomic profiling. * off-label therapy.

## Data Availability

The data used to support the findings of this project are available from the corresponding author upon request.

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
