# Peer review of "The Challenges and Opportunities of the Implementation of Comprehensive Genomic Profiling in Everyday Clinical Practice with Non-Small Cell Lung Cancer: National Results from Croatia"

_cancers, 2023, doi:10.3390/cancers15133395_

Round 1
Reviewer 1 Report
The main question the authors address in their study is the necessity of an implementation of comprehensive genomic profiling in everyday clinical practice in patients having non-small cell lung cancer. In their publication authors present, although on a relatively small number of patients what is a consequence of restrictions imposed by the Croatian regulatory and insurance company (reference 12), obtained results. Topic is relevant for the oncologists dealing with the patients having lung cancer. It adds that comprehensive genomic profiling should replace everyday practice which is based on analysis of only several genomic alterations. Moreover, it is also a more rational method or approach for testing, especially in case of lung cancers where usually there is a limited amount of tumor tissue available. I do not have objections on how the manuscript is written, including the part with the discussion and conclusion section. References are appropriate but are inconsistently written (for example, some start with the first name and not by the author's family name).
References are inconsistently written! Present them consistently.
Whenever available "doi" should be included.
Reference 12 is not complete. Add where it is published or add internet availability.
Author Response
Reviewer #1
The main question the authors address in their study is the necessity of an implementation of comprehensive genomic profiling in everyday clinical practice in patients having non-small cell lung cancer. In their publication authors present, although on a relatively small number of patients what is a consequence of restrictions imposed by the Croatian regulatory and insurance company (reference 12), obtained results. Topic is relevant for the oncologists dealing with the patients having lung cancer. It adds that comprehensive genomic profiling should replace everyday practice which is based on analysis of only several genomic alterations. Moreover, it is also a more rational method or approach for testing, especially in case of lung cancers where usually there is a limited amount of tumor tissue available. I do not have objections on how the manuscript is written, including the part with the discussion and conclusion section. References are appropriate but are inconsistently written (for example, some start with the first name and not by the author's family name).
[1] References are inconsistently written! Present them consistently.
Response: Thank you for your dedicated time and revision. We have carefully analyzed the references and have changed them according to the Cancers journal requirements.
[2] Whenever available "doi" should be included.
Response: Thank you for your constructive advice and DOI has been added in all references where available.
[3] Reference 12 is not complete. Add where it is published or add internet availability.
Response: Thank you pointing this. The reference 12 has been updated, line 399-400.
Now, it states:
“14. Babić D, Pleština S, Samaržija M, Tomić S, Vrdoljak E, Ban M, Belac Lovasić I, Belev B, Ćorić M, Dabelić N, et al., Preporuke Za Odabir Bolesnika/tumora Za SGP (2021). Availableat: http://www.hrvatsko-onkolosko-drustvo.com/wp-content/uploads/2021/02/Preporuke-za-SGP_Izdanje-23.2.2021.pdf. Accessed: June 25, 2023.”

Reviewer 2 Report
I have read and reviewed the manuscript “Challenges and opportunities of the implementation of comprehensive genomic profiling in everyday clinical practice of non-small cell lung cancer: national results from Croatia”. The authors do a nice job of a contemporary review of the use of precision medicine in their country. They also highlight the pros and cons. It is interesting that in Croatia testing permits patients to know all their mutations, but insurance and reimbursement then shifts the burden to the patient about half of the time by not covering. This is an ethical dilemma, the authors state that while knowing the biomarkers is a “basic human right”, how is providing appropriate reimbursed care not the standard that should be obtained versus leaving it to the patient to “find a way to receive optimal treatment for their disease”? I recognize this may be beyond the scope of the paper, but the tone of the manuscript at times is not patient-centric.
Major Points:
1. Since it appears that this study was conducted in six hospitals in Croatia, it would be of value to the reader to have a better understanding of the scope of lung cancer in Croatia. Could a brief summary be added to the INTRODUCTION? This would help put the 48 patients in this study into context. This might include the burden of disease annually in Croatia. An abstract from a few years ago in Journal of Thoracic Oncology details epidemiological lung cancer data in Croatia. This might be a valuable reference.
2. Is it possible for the authors to use their results to suggest a “slimmed down” panel to be tested that could more easily be done locally without the need for sending specimens to FoundationOne?
3. While the authors state some pros of liquid biopsy, there are certainly negatives such as the result being negative after treatment despite viable tumor still being present. A slightly more balanced statement would likely be of benefit.
4. Based upon the IMPower010 trial, PD-L1 drugs are now being given for any tumor with greater than 1% expression. Therefore, would the authors consider NOT performing PD-L1 testing and TMB testing, but only one of them? Also, since 23 of 30 patients (77%) had a PD-L1 testing, the authors may want to consider highlighting this point when they make statements that 96% of patients had any genomic alteration and 79% were clinically relevant.
5. Why do the authors consider KRAS a “clinically relevant” alteration? Are they recommending Sotorasib for their patients? This drug is most effective in G12C patients, what were the specific KRAS alterations in the 16 KRAS positive patients?
6. When the authors state that the median number of clinically relevant genomic alterations was 2, does this mean that the patient had all their genomic alterations being clinically relevant (i.e. targetable with drug therapy)? How often were the additional alterations either PD-L1 or KRAS?
Minor Points:
1. The first time that an abbreviation is used it should be written out. CGP is not defined in the manuscript as comprehensive genomic profiling initially, but eventually.
2. Missing a bracket after reference 12 in the INTRODUCTION.
3. Were all the samples sent to FoundationOne for analysis? This is unclear in the manuscript as it states “it was done through FoundationOne”.
4. Do the authors really mean “apostrophized”? Could this be a typo?
Overall English is quite good, there are a few minor issues.
Author Response
Reviewer #2
I have read and reviewed the manuscript “Challenges and opportunities of the implementation of comprehensive genomic profiling in everyday clinical practice of non-small cell lung cancer: national results from Croatia”. The authors do a nice job of a contemporary review of the use of precision medicine in their country. They also highlight the pros and cons. It is interesting that in Croatia testing permits patients to know all their mutations, but insurance and reimbursement then shifts the burden to the patient about half of the time by not covering. This is an ethical dilemma, the authors state that while knowing the biomarkers is a “basic human right”, how is providing appropriate reimbursed care not the standard that should be obtained versus leaving it to the patient to “find a way to receive optimal treatment for their disease”? I recognize this may be beyond the scope of the paper, but the tone of the manuscript at times is not patient-centric.
Major Points:
[1] Since it appears that this study was conducted in six hospitals in Croatia, it would be of value to the reader to have a better understanding of the scope of lung cancer in Croatia. Could a brief summary be added to the INTRODUCTION? This would help put the 48 patients in this study into context. This might include the burden of disease annually in Croatia. An abstract from a few years ago in Journal of Thoracic Oncology details epidemiological lung cancer data in Croatia. This might be a valuable reference.
Response: Thank you for your dedicated time and revision. We absolutely agree that in order to have better understanding of the scope of lung cancer in Croatia, we must present epidemiological details. So, we have put the incidence and other data in the introduction with new reference from our Cancer register, lines 58-61, reference line 380.
Now, the part of introduction states:
“In Croatia, the burden of lung cancer is rather large with approximately 3000 newly diagnosed patients per year, making it second most common both males and females with mortality to incidence ratio of 0.95 for males and 0.85 for females [2].
Reference:
- Hrvatski zavod za javno zdravstvo, Registar za rak Republike Hrvatske. Incidencija raka u Hrvatskoj 2020., Bilten 45, Zagreb, 2022.“)
[2] Is it possible for the authors to use their results to suggest a “slimmed down” panel to be tested that could more easily be done locally without the need for sending specimens to FoundationOne?
Response: Thank you for this valuable comment. We are in the process of opening our, Croatian based comprehensive genomic laboratory that will be used for an all-large panel-based genome tastings and, of course, will use different multigene platforms.
[3] While the authors state some pros of liquid biopsy, there are certainly negatives such as the result being negative after treatment despite viable tumor still being present. A slightly more balanced statement would likely be of benefit.
Response: Thank you for your constructive inputs. We have tried to balanced more the statement about liquid testing in the introduction, lines 92-93.
Now, the part of introduction states:
(„ However, negative findings of liquid testing do not exclude presence of viable tumor with potentially targetable mutations.“
Reference:
- Li, B T, F Janku, B Jung, C Hou, K Madwani, R Alden, P Razavi, J S Reis-Filho, R Shen, J M Isbell, et al., “Ultra-deep next-generation sequencing of plasma cell-free DNA in patients with advanced lung cancers: results from the Actionable Genome Consortium.” Annals of oncology : official journal of the European Society for Medical Oncology vol. 30,4 (2019): 597-603. doi:10.1093/annonc/mdz046)
[4] Based upon the IMPower010 trial, PD-L1 drugs are now being given for any tumor with greater than 1% expression. Therefore, would the authors consider NOT performing PD-L1 testing and TMB testing, but only one of them? Also, since 23 of 30 patients (77%) had a PD-L1 testing, the authors may want to consider highlighting this point when they make statements that 96% of patients had any genomic alteration and 79% were clinically relevant.
Response: Thank you for this nice comment. From the everyday oncology practice point of view, we have to do the PD-L1 testing in order to potentially apply the immunotherapy treatment. The tumor mutational burden is the standard part of the FoundationOne panel and it is done automatically by them (could not be changed by us).
Yes, we have highlighted the importance of size of PD-L1 positive population, adding it to the text of results, lines 228-229.
Now, the part of introduction states:
(“PD-L1 status was determined for 30 (63%) patients, out of which clinically positive results were reported for 23 (77%) patients.“)
[5] Why do the authors consider KRAS a “clinically relevant” alteration? Are they recommending Sotorasib for their patients? This drug is most effective in G12C patients, what were the specific KRAS alterations in the 16 KRAS positive patients?
Response: Thank you for this valuable question. The “clinical relevance” is defined by the Foundation One report basing it on a large database of biomarkers and not by us, we have explained its meaning in the Methods, line 166-168. That is why Sotorasib is not recommended for all KRAS alterations, it is only mentioned as potentially effective drug but still in phase of clinical trial. The reason why we have pointed out only the percentage of G12C mutations is because we can administrate to our patients Sotorasib from NPP programs. While, all KRAS alterations are stated in the sheet 2 of supplementary file.
[6] When the authors state that the median number of clinically relevant genomic alterations was 2, does this mean that the patient had all their genomic alterations being clinically relevant (i.e. targetable with drug therapy)? How often were the additional alterations either PD-L1 or KRAS?
Response: Thank you for comment and detailed revision. As previously stated, clinically relevant genomic alteration implies either on-label or off-label targeted therapy or potential clinical trial, lines 166-168. Hence, some of the clinically relevant alterations did not have targetable drug opted but only mentioned its potential effect in clinical trial. We have calculated the median number of clinically relevant genomic alterations including KRAS alterations in all patients. However, we did not include expression of PD-L1 which is performed by immunohystochemistry and not by CGP.
Minor Points:
[1] The first time that an abbreviation is used it should be written out. CGP is not defined in the manuscript as comprehensive genomic profiling initially, but eventually.
Response: Thank you for this valuable observation. We have corrected it and have checked all other abbreviations, line 30.
Now, the part of simple summary states:
(“Consequently, the introduction of comprehensive genomic profiling (CGP) in everyday diagnostic and clinical practice is one of the imperatives that could benefit everybody involved.”)
[2] Missing a bracket after reference 12 in the INTRODUCTION.
Response: Thank you once again for very valuable observation. We have added a bracket.
[3] Were all the samples sent to FoundationOne for analysis? This is unclear in the manuscript as it states “it was done through FoundationOne”.
Response: Thank you for this comment. Yes, all samples were sent for analysis, we have tried to point it out in order to be more understanding; in the Methods, lines 117-120.
Now, the part of methods states:
„The analysis of tumor tissue or blood samples was performed through FoundationOneCDx or FoundationOneLiquidCDx for all patients in a certified Clinical Laboratory Improvement Amendments, College of American Pathologists accredited laboratory (Foundation Medicine Inc., FMI, Cambridge, MA, USA) [25-27].“
[4] Do the authors really mean “apostrophized”? Could this be a typo?
Response: Thank you for this feedback. We have changed in into emphasized to make it more clear, conclusion, line 323.
Now, the part of conclusion states:
„Thus, we are facing the same issues regarding NSCLC management as previously emphasized.“
